# Exploring ‘Wether’ Grazing Patterns Differed in Native or Introduced Pastures in the Monaro Region of Australia

**DOI:** 10.3390/ani13091500

**Published:** 2023-04-28

**Authors:** Danica Parnell, Jack Edwards, Lachlan Ingram

**Affiliations:** 1The School of Life and Environmental Sciences, Faculty of Science, The University of Sydney, Sydney, NSW 2006, Australialachy.ingram@dpi.nsw.gov.au (L.I.); 2NSW Department of Primary Industries, Orange, NSW 2800, Australia

**Keywords:** grazing behavior, sheep, pasture management, small ruminant management

## Abstract

**Simple Summary:**

Understanding how and why livestock choose to graze on pasture can provide valuable information to farmers, such as overall pasture use, foraging patterns, and behavior trends. This information may be used to further assist farmers in making management decisions on how to increase production. This study specifically aimed to understand what the important drivers of grazing were, such as landscape attributes (pasture quality, distance to trees, distance to fences, distance to water) and climatic variables (rainfall, temperature), in paddocks comprising either native or improved/introduced grass species, in the Monaro region of SE Australia. Pasture quality was a highly important and a significant variable, along with temperature and elevation for the improved pasture paddock, in contrast to the native paddock, where in addition to elevation and temperature, near distance to trees was more important in determining sheep location. Results from research such as this can provide valuable insights into various grazing behaviors of sheep and also assist in shaping the sustainable management of sheep.

**Abstract:**

Monitoring livestock allows insights to graziers on valuable information such as spatial distribution, foraging patterns, and animal behavior, which can significantly improve the management of livestock for optimal production. This study aimed to understand what potential variables are significant for predicting where sheep spent the most time in native (NP) and improved (IP) paddocks. Wethers (castrated male sheep) were tracked using Global Positioning System (GPS) collars on 15 sheep in the IP and 15 in the NP, respectively, on a property located in the Monaro region of Southern New South Wales, Australia. Trials were performed over four six-day periods in April, July, and November of 2014 and March in 2015. Data were analyzed to understand various trends that may have occurred during different seasons, using random forest models (RFMs). Of the factors investigated, Normalized Difference Vegetation Index (NDVI) was significant (*p* < 0.01) and highly important for wethers in the IP, but not the NP, suggesting that quality of pasture was key for wethers in the IP. Elevation, temperature, and near distance to trees were important and significant for predicting residency of wethers in the IP, as well as the NP. The result of this study highlights the ability of predictive models to provide insights on behavior-based modelling of GPS data and further enhance current knowledge about location-based choices of sheep on paddocks.

## 1. Introduction

With advancements in tracking technology for livestock, extensive animal production systems have the ability to compile uninterrupted data over a range of environmental conditions and allow us to better understand both foraging patterns/behaviors and spatial distribution of animals [1]. Combining this technology simultaneously with pasture-based data, insights can be gleaned that can provide a better understanding of paddock utilization and behavioral interactions [2,3,4]. Access to these technologies is more readily available and can reduce the need for observational data, which are quite intensive in nature due to the demand on labor and range of hours required [5,6,7,8]. As such, detailed behavioral studies with this technology can assist in further refining the way sheep are managed, with increased knowledge being available on grazing distribution patterns and the influence of various behavioral attributes. For example, social interactions are known to have a large influence on how sheep graze [9]. Sheep may choose to graze non-preferred pasture species to stay close together [9,10,11]. When in larger herd sizes, sheep have been reported to form subgroups of up to five or six sheep [12] or to leave the main flock if they are followed by other animals [10,13]. Furthermore, there are reported individual personalities which sheep can exhibit that will overall influence grazing choices (i.e., choosing poorer forage and revisiting areas versus breaking off from the herd for better-quality forage) [13].

Aside from the aforementioned livestock factors, environmental factors are also known to influence grazing behaviors, such as time of day and climate. Sheep have been reported to graze just prior to daybreak, often camp during the middle of the day, then graze once again in the late afternoon [12,14,15]. This grazing pattern is also often referred to as diurnal grazing behavior [9]. The grazing around the dusk period that occurs often has a longer observed grazing window. It has been hypothesized that the reason for this is that sheep consume forage, which is higher in palatability to fill the rumen for longer overnight [16,17,18]. Weather variables such as temperature may also have an impact on grazing behaviors. Ewes were reported to visit watering points and have reduced grazing times when temperatures were higher, whereas on cooler days, they grazed more frequently [19]. Wethers in contrast were found to spend more time grouped together at lower temperatures and were less grouped when temperatures reached above 25 °C on high stocking rates [9]. Pasture availability is another known variable which influences grazing patterns exhibited. Sheep may selectively graze areas in paddocks purely based on the forage availability, with preference for species such as legumes, until that particular feed is either restricted or limited [3,12,17,18].

Pasture systems in Australia are composed of native, improved, or mixed pasture systems, with native pastures being more predominant than the others [20]. Though native pastures have the ability to provide reliable feed during all seasons due to their hardiness and ability to thrive when water and nutrients are limited, improved pastures offer higher quality and quantity of feed [21,22]. Improved pastures have higher levels of productivity and better responses to fertilization, which largely contrasts with lower-production native pastures found on poorer-quality soils that often receive lower rates of fertilizer [21,22]. Native grasslands often contain a higher proportion of C4 (summer active) grasses, which are usually of lower quality and produce less biomass than the C3 (spring active) grasses that are more commonly found in improved pastures. These C3 grasses have maximal growth rates in spring and potentially a smaller amount of growth in autumn and late spring; however, they can also be largely impacted by climate and weather. Therefore, though improved pastures are the preferred pasture option due to the larger quantity and greater quality of forage they provide, they must be effectively managed in these cultivated lands for livestock to reap the benefits.

The overall objective of this study was to determine which variables were best for predicting sheep location in paddocks of native and improved pastures in the Monaro region using global positioning system (GPS) data. In addition, both observational data and accelerometer data were used for additional information to the GPS data to predict grazing periods throughout the day. Our hypothesis is that animals here will seek out areas with higher Normalized Difference Vegetation Index (NDVI) values regardless of the pasture type (evident through higher residency and importance value). It is also assumed that wethers may spend more time grazing in the native pastures than the improved pastures due to the quality disparity between both pasture types. Pasture quality may therein play a larger role in grazing behavior than environmental and social behaviors. With a better understanding on how these interactions occur, this research could not only aid in better management of both sheep and pasture in the Monaro region of Australia, but also improve current knowledge about prediction-based modelling for grazing using GPS data.

## 2. Materials and Methods

This research was performed on a Merino wool-based enterprise, ‘Coolringdon’ (36.28° S, 148.98° E), in the Monaro region of southern NSW. Research was conducted over four six-day periods in April (T1; autumn), July (T2; winter), and November (T3; spring) of 2014 and March (T4; summer/autumn) of 2015, with the aim to encapsulate behavior across all seasons. The Monaro region is often characterized by low ambient temperatures, high wind chill, and severe frosts, all which are common from April to September. Annually, the region receives on average 540 mm of rain, with a summer dominant pattern in 2014 [23]. Prior to the initiation of this study, there was a long summer drought, which was followed by heavier-than-usual rainfall during the study (Figure 1). It is important to highlight, however, that due to this unusual pattern of rainfall in 2014, pasture growth was much greater over the duration of the study than what may be normally expected.

Two paddocks were chosen that were within close proximity (<100 m) of each other (Figure 2) and had soil profiles that matched their pasture composition. The two main soil profiles of this area are basalt and granite originated from basalt and granite, with basalt-derived soils being more productive and used for improved pastures and the granite-derived soils being less productive and used for native pastures [22]. The Monaro region of Australia is known to have one of the largest native pasture systems across all of New South Wales (NSW), at 70% [22], and as such, native grasslands are often an important component of sheep grazing operations in the area. The “native” paddock (NP) (21.3 ha^−1^) was mainly composed of *Hordeum leporinum, Poa* spp., and *Stipa* spp. This paddock had a smattering of *Eucalyptus* spp. trees in the northern end and had an elevation of 984 m above sea level (asl) in the north and 935 m in the southern end and had a water trough situated in the southwestern corner (Figure 2). The “improved” paddock (IP) (26.6 ha^−1^) was composed of *Festuca* spp., *Phalaris aquatica*, and *Trifolum subterraneum*. Unlike NP, the IP had relatively sparse shade coverage due to the lack of trees present overall, had an elevation range of 932–949 m, and had a water trough located in the northwestern area of the paddock (Figure 2). The IP in particular, however, had a row of *Pinus radiata* which often provided protection from prevailing winds. In both paddocks, the positions of trees and water troughs were logged using a handheld GPS unit (Garmin etrex 30, Garmin Ltd., Olathe, KS, USA).

It is important to note that during T1, T2, and T4, it was not possible to perform a botanical survey, as grasses were within a vegetative state and therefore, there was an inability to identify all pasture species in both paddocks. Paddocks were sampled for their normalized difference vegetation index (NDVI) 1–2 days prior to the initiation of each trial, on approximately 40 m transects across each paddock. NDVI within the study is utilized as a proxy for forage quality and quantity across the landscape [24], as the data obtained from the collection are collected at a relatively fine scale. The NDVI values of each paddock were measured using a vehicle-mounted ACS-470 CropCircle System (Holland Scientific, Lincoln, NE, USA) using the red band wavelength (670 nm) and the near-infrared band wavelength (760 nm) at a rate of 20 Hz, with GPS coordinates continuously recorded alongside it using a Trimble 250 EziGuide (Trimble Navigation Ltd., Westminster, CO, USA). All data were stored on a data logger before being processed using Variogram Estimation and Spatial Prediction with Error (VESPER) software [25] to produce an interpolated 5 × 5 m gridded NDVI map, which was then imported into ArcGIS (10.5 ESRI, Redlands, CA, USA) for analysis (Figure 3). The average NDVI for each paddock was relatively consistent with values of 0.273 ± 0.025 for T1 (Figure 3a), 0.217 ± 0.036 for T2 (Figure 3b), 0.235 ± 0.035 for T3 (Figure 3c), and 0.222 ± 0.015 for T4 (Figure 3d).

Wethers were chosen from the commercial flock of ‘Coolringdon’, and for the course of the trials, they were maintained as a separate mob from other sheep present on the property and were subject to the same standard farming operations (drenching, shearing, etc.). Fifty wethers approximately three years old were utilized for the study as approved by the University of Sydney University Animal Ethics Committee (2014/570). Of the 50 wethers chosen, 25 were randomly allocated into the IP, and the remaining 25 were allocated to the NP. Wethers were rotated between the two paddocks for each trial, for example, wethers in NP for T1 were assigned to the IP for T2, then back to the NP for T3, and so on. Of the 25 sheep in each paddock, GPS collars were randomly allocated to 15 sheep in each paddock to record their locations (‘fixes’) over a minute at 0, 15, 30, 45, and 60 s, before going into sleep mode for 4 min. The remaining 10 animals acted as buffer sheep. Logging of GPS data occurred from the day before each trial started to around lunchtime of the final day of the trial. Once the collars were removed from the sheep on the 7th day, the data were cleaned to remove any fixes that took longer than 16 s to log, as it was not an accurate representation of location. Data were imported into ArcGIS and any points which were outside paddock boundaries were also removed.

Over the course of the four trials, various sheep behaviors were visually observed to understand how often various traits occurred. At the same time that GPS collars were placed on the randomly selected 15 sheep in each paddock, a numbered, colored vest was also placed on the wethers to identify the individuals. The traits recorded were standing (S), standing grazing, walking grazing, standing walking grazing, lying (L), walking, running, and drinking. Due to the low levels of recordings, walking and running were combined into ‘moving’ (M); standing grazing, walking grazing, and standing walking grazing were combined into ‘grazing’ (G); and as only a total of four observations of drinking was recorded across all the trials, it was removed from the analysis. Measurements started around dawn and finished at dusk, occurring every two hours, lasting for six hours across the day. Starting on the hour, the observed behaviors of each sheep were recorded until all sheep had a recorded observation, using a scan sampling technique [26,27]. At 10 min past the hour, the process was repeated and occurred every 10 min. In total, observations were recorded in 10 min blocks over an hour, until all 15 bibbed wethers had recorded observations. As there was only one observer throughout the trials for consistency of observations, sheep on the NP were observed on days one, three, and five, whilst sheep on the IP were observed on days two, four, and six.

IceTags are a three-axis accelerometer that was attached to the leg of a sheep and collected count data of steps and lying bouts to ultimately provide a count of the time spent lying and standing over one-minute intervals. IceTags were placed on the legs of seven randomly selected wethers per paddock during the collar- and cloak-fitting process prior to the start of the trials. As a smaller number of IceTags were available than GPS collars (14 IceTag accelerometers, 30 GPS collars) and there was a loss of units due to battery failure, seven sheep per paddock were fitted with the unit in T1 and six sheep per paddock in T2.

Speed was calculated from the GPS data using both the date and time stamp to obtain a ‘seconds’ (time) value and a distance between the recorded coordinates (easting and northings) for each “fix” of the animal at the time. Distance between fixes (*x*) is a measure of the difference between two points and was calculated as follows:(1)$$x=\sqrt{\frac{{\left(aa-a\right)}^{2}}{{\left(bb-b\right)}^{2}}}$$
whereby *aa* is the second eastings value, *a* is the first eastings value, *bb* is the second northings value, and *b* is the first northings value. Once the distance (m) by fixes is obtained, speed (m/s) was calculated by dividing the distance by the seconds.

Data were analyzed for each of the trials separately to compare the various trends that may have occurred within each season. The objectives of each model were to understand what variables were best for determining where sheep were located in a paddock (IP, NP) across each season (T1, T2, T3, and T4). Sheep location was based on a livestock residency index (LRI), a count of the number of times an animal spent in a given pixel (5 × 5 m) per hour, calculated for each individual animal and each treatment [4,27,28]. The LRI was calculated for each treatment (*n* = 15 collared animals; treatment replicate LRI) and was the target variable for all models generated.

Random forest modelling (RFM) was used to explain and predict sheep location and has well-established means to explore spatially mapped data in agriculture and conservation studies [29,30,31,32]. During the RFM process, data are internally separated into a validation (~33%) and training (~66%) to form an out-of-bag (OOB) subset. Details of how RFMs process data have been outlined previously [33,34]. Mean squared error (MSE) and R^2^ values taken from these OOB subsets were utilized as an indication of overall performance of each model, importance values were used to identify the scale and relative importance of each variable in the model prediction, and partial dependence plots (PDPs) were used as a visualization tool depicting the effect of the selected predictor variable (LRI) on the response variable.

The data prepared for RFMs included near distance to various features such as trees (NTs), water troughs (NWs), and fence lines (NFs), as well as elevation (EL), aspect (AS) and the normalized difference vegetation index (NDVI) value as a measure of pasture quality [35]. Additionally, weather data were included to understand the importance of climatic variables such as rainfall (mm) and cumulative rainfall (mm), temperature (°C), apparent temperature (°C), dew point, relative humidity (%), wind direction, wind speed (km/h and m/s), wind gust (km/h), and wind pressure (QNH, MSL), sourced from the Cooma Airport weather station [23]. Data for each animal (A) were also included to identify if there were any significant influences by individual animal behavior [9]. Due to high collinearity between the climatic data, variables with high variance inflation factor (VIF) values over five were removed, leaving rainfall (R) and temperature (T). Sheep wind chill (WC) was calculated based on calculations determined by Weeks et al. [36]. Additionally, eastings (EA) and northings (NO) were also included in the models to attempt to reduce the potential for biased random forest models and account for any potential spatial autocorrelation in the GPS data. Data for each individual animal were also included in the models to determine if an individual animal’s behavior was a significant factor on grazing choices [9]. In total, 12 variables were included as predictors in the final IP and NP models. Data were analyzed using packages ‘ranger’ (V0.12.1), ‘pdp’ (V0.7.0), ‘car’ (V3.0-12) [37,38,39] using R 4.03 [40].

Models were produced for ‘all hours’, which aimed to identify important variables for predicting where animals would be located across a diurnal cycle, and models were also produced for ‘grazing hours’ (GH), which were used to identify key drivers of why sheep were in a given location during times when grazing was observed. Using the aforementioned visual observations of grazing behavior, the GH model only included the times in which grazing observations were recorded. In this paper, the smallest-sized dataset belonged to the GH model for T4 NP, with *n* = 9742 for calibration and *n* = 4799 for validation, and the largest dataset belonged to the ‘all hours’ (AH) model for T3 IP, with *n* = 70,368 for calibration and *n* = 34,659 for validation.

## 3. Results

### 3.1. Movement Data

The predominant behavior observed across all seasons was grazing (45% for IP, 41% for NP), except for in T3 where the dominant behavior for IP was lying (38%) followed by grazing (29%), which contrasts from the NP, where lying and standing were equally proportionate (32%) (data not shown). From these observations, the grazing activity counts were taken, and plots were produced for each trial and treatment (Figure 4). There was variation with grazing counts for T1–T3 in both paddocks across each hour, but in contrast, there was consistency across the hours for T4 in the IP and NP (except during the fourth hour) (Figure 4).

After calculating the average speed (m/s) of wethers across the different seasons, distinct patterns were evident for each trial and per treatment basis (Figure 5). Sheep in the IP consistently spent more time walking across the course of the day, in contrast to the NP, which had more uniform ‘patterns’ of higher speeds/activity during the morning and in the afternoons (Figure 5). Both treatments spent greater amounts of time grazing throughout the day during T2, as well as the NP in T1 (Figure 5a,b). Speeds were often greater than 0.15 m/s in the mornings and afternoons for both paddocks, except for in T2, where wethers in the IP spent more time walking at a faster pace than 0.15 m/s regardless of the time day.

IceTag accelerometer data showed an inverse relationship between step count and lying-bout counts (Figure 6). During the early hours of the morning and the hours in the late afternoon when step count decreased, lying count increased for both treatments (Figure 6). Across both treatments and trials, there seemed to be a trend roughly around the first few hours of daylight (~0600 to 0900) and the last few hours in the afternoon before nightfall (~1500 to 1700) where the most steps were taken by wethers (Figure 6a,b). Furthermore, fewer steps were taken overall across the day by the wethers in the NP, in contrast to those in the IP, and this is more apparent in T1 than T2 (Figure 6a,b).

### 3.2. Random Forest Models

Models were created for each trial to gain an understanding of the various seasonal effects that occurred across the duration of the study. The best performing model overall was T1 IP grazing hours (0.98 R^2^; 2.0 × 10^−6^ MSE), and the lowest performing model was T3 NP all hours (0.94 R^2^; 3.0 × 10^−6^ MSE) (Table 1). Across all of the trials, the significance (*p* < 0.01) and importance values varied for each of the variables included, however, distinct similarities were also evident. As the main intention is to identify variables which may drive grazing behaviors, the next section will predominantly focus on the results from the GH models, and where relevant, include a brief discussion of the AH models. General trends for all of the models produced often saw A, R, and NF as the three least important variables, regardless of the treatment or trial (Table 2). Similarly, EA, NO, and NW were commonly ranked as the top three important variables, except for certain periods where they were ranked between fourth and seventh (Table 2) and appeared to be influenced by the season more in the IP than NP.

In addition, residency (LRI) maps were created to visually identify the predominant locations where sheep were spending time during each of the trials (Figure 7). Evidently, for each trial, there were areas of the paddocks that were completely unused, predominantly in the NP compared to the IP (Figure 7). Wethers in the NP clearly congregated within the northernmost part of the paddock, where there were large amounts of tree cover (Figure 2) and there was the highest elevation across all trials. In contrast, LRI varied, as during each of the different trials for the IP, as in T1 and T4, wethers mainly resided on the southeastern area of the paddock (Figure 7a,d), whilst in T2, wethers resided closer to the tree line in the northwestern area of the paddock (Figure 7b). Interestingly, wethers had consistent residency levels across the entire paddock in IP for T3 (Figure 7c).

#### 3.2.1. Improved Paddocks

The IP models produced for grazing hours for each of the trials produced an OOB R^2^ range from 0.95 (T2) to 0.98 (T1), with MSE values between 2.0 × 10^−6^ (T3) and 3.0 × 10^−6^ (T2) (Table 1). Interestingly, based on importance value, NDVI was the second highest ranked variable for the IP, regardless of hours included in the model, for T1 only, and only became important again in T3 during GH (Table 2). NDVI was also significant for all trials except for T2. Weather variables such as T changed in their relative importance depending on the season; T was ranked third and second only in T1 and T2 for both GH and WC, which, for the IP, was only ranked highly during T2 for both AH and GH and T3 for only GH (Table 2). Each of the variables and their rankings, as aforementioned, were also significant (*p* < 0.01). The landscape variable AS was not significant or as important compared to other variables for the IP. However, in contrast, EL was ranked fifth during GH for T1 and T4, despite its statistical insignificance (Table 2). NT was only moderately important, often ranking between sixth and eighth during GH; however, it was placed as the third and fifth most important variable for T4 and T3 AH (Table 2). Significance only occurred during T1 and T3 GH and T3 and 4 AH. The landscape variable NW was within the top three important variables for all trials except for T1 and was only significant for GH in T3 (Table 2). Furthermore, EA was often ranked highly for all trials but was only significant in both AH and GH for T3 and T4. In contrast, NO was ranked first in T1, T4, both AH and GH, in T2 AH, and was significant for GH in T1 and T4, but only T2 and T4 AH (Table 2).

Partial dependence plots produced showed the general trends between each of the variables and residency by wethers. For GH in all trials, as NDVI values increased, the LRI by wethers had decreased, often occurring when NDVI reached 0.2, except for in T1 where this was when NDVI reached 0.25 (Figure A1). General trends for both EA and NO saw that as they both increased, so too did residency, often suggesting that wethers in the IP spent time around the northwestern corner of the paddock (Figure A1 and Figure 2). Landscape variable EL saw a similar trend where regardless of the trial, as EL increased to 930 m ASL, so too did LRI (Figure A1). Interestingly, AS had the highest residency when south-facing (between 150° and 200°) for all trials except for T2, where LRI was greatest when AS was western-facing (between 250° and 300°) (Figure A1). Residency was highest for NF when wethers were between 0 and 2 m from a fence line, regardless of the trial. Contrastingly, NT saw variations in residency depending on the trial, as for T1, residency was highest when wethers were more than 100 m from trees but was higher when animals were within the first 100 m from a tree for T2–T4 (Figure A1). In addition, NW was another variable which fluctuated with LRI depending on the trial, as within T2 and T3, LRI was greatest when animals were 600 m or more away from water troughs; however, in T4, predicted residency was greatest when wethers were within 50 m or at least 550 m away from a water trough (Figure A1). As expected, all weather variables changed predicted LRI depending on the trial/season. As seen in WC, higher residency was evident when WC was lower (500–700) for T3 and T4; however, in T1 and T2, the predicted LRI was higher as WC was elevated (800 for T1, 750 and 900 for T2). This could have been correlated with the effects of T, as residency was often higher at lower temperatures during T3 and T4 but was higher as T increased (15 °C or greater for T1 and 5 °C for T2) for T1 and T2 (Figure A1). Trends between plots created for GH were similar for AH.

#### 3.2.2. Native Paddocks

Models created for the NP produced an OOB R^2^ range of 0.94 (T3) to 0.96 (T1) and MSE values of 3.0 × 10^−6^ and 4.0 × 10^−6^ (Table 1). The landscape variable EL was one of the higher-ranked variables across all trials, being significant for all except T2 (Table 2). In contrast, while the importance value for AS often ranked quite low across all trials, nonetheless they were generally significant for the GH models, except for T2, and all trials for AH (Table 2). Contrastingly, NT was ranked moderately, being significant for GH in all seasons except for T2 and only T3 AH (Table 2). Weather variables were significant for all trials during GH, with T being ranked moderately high, R being ranked quite low, and WC being ranked moderately high, except for in T2 where it was ranked first for GH or fourth for AH (Table 2). Of interest, NDVI was not a highly important variable to wethers in the NP as much as it was for those in the IP (Table 2). NDVI was, however, significant for all trials except T2 for GH and was only significant in T1 for AH in the NP (Table 2).

Across all trials, sheep were generally located either well away (>400 m) from trees, at often the southern end of the paddock where pasture quality was higher (Figure A2 and Figure 3), or were relatively close to trees (<50 m), regardless of AH or GH, likely indicating that trees were often providing shelter to the sheep from factors such as sun, rain, and wind (Figure A2). In contrast to NT, NF had higher LRI values only when animals were between 0 and 4 m away from fences, regardless of the season (Figure A2). Similarly, partial dependence plots (PDPs) for NW across all trials had low LRI until wethers were at least 600 m away from a water trough (Figure A2). Due to sheep spending the majority of their time at the northern and higher end (>960 m ASL) of the paddock, EL was evidently an important driver of sheep location. Similarly, as the paddock was southern-facing, AS had higher residency in T1 and T2 when sheep were southeasterly (between 100° and 200°) and western-facing (above 200°) in T3 and T4 (Figure A2). The level of residency for NDVI varied across the trials; however, the greatest LRI often occurred when NDVI values were around the 0.2–0.3 mark, regardless of higher NDVI values available (Figure A2). The weather variable WC saw greater LRI as WC increased for T1–T3, but lower residency as WC increased in T4. Interestingly, residency dramatically increased for WC in T2 (0.05, compared to 0.02 in T1, 3, and 4) when wind chill was 900 or greater (Figure A2). Rainfall also decreased residency in wethers as R increased, except for in T2 where LRI was also high when there was 3 or more mm of R (Figure A2). Residency by wethers also varied depending on T and generally saw a decrease in residency as T increased for T1 and 3, but there was greater grouping as T increased for T2 and 4 (Figure A2). Additionally, trends for NO saw higher LRI values when NO was greater in all trials and suggested a relatively even distribution of LRI for EA in T1, and high residency occurring at either the most western or eastern point (lower or higher value) of the paddock for T2–4 (Figure A2). The forecasted residency that occurred between AH and GH was quite similar, except for in T2 where residency within the first 100 m from a tree was greater than further away, and higher residency was also evident at lower NDVI values (0.10–0.20).

### 3.3. Weight Data

Weight data collected over the course of the study show that sheep in the IP increased in their liveweights, whilst sheep in the NP lost weight across the duration of the study. There was no effect of trial or trial x treatment on net weight changes; however, there was a pure treatment effect with an average of 1 kg difference between IP and NP. Wethers in the IP gained ~0.2 kg per trial, whilst those in the NP lost ~0.8 kg.

## 4. Discussion

### 4.1. Movement Behavior Data

All of the ‘movement’-based data evidently had similar trends to each other and provided a sound basis for estimating when grazing was likely to occur. The behavioral observations presented in Figure 4 had similar trends with calculated speed data (Figure 5) and the IceTag data presented for T1 and T2 (Figure 6), as there were distinct peaks for T1–T3 for both treatments, both in the morning and the early afternoon. Evidently, trends in T4 were likely apparent that as more pasture became available, sheep were grazing during each observation period. However, this inconsistency could also be attributed to a change in observers that occurred for T4. As has been highlighted in existing literature, a change in observers is likely to cause inconsistencies between results, attributed to human-based errors [41]. Visual monitoring and observation of animals in research is often of a labor-intensive nature, and as such there is a demand for alternative methods of observing animals [41]. Aside from this, sheep are likely to be easily startled during observation periods if the observer makes sudden movements, impacting how accurate an observation period may be. The ability to monitor the behavior of certain livestock without requiring the labor-intensive nature of visual observations would therefore be useful for research. Furthermore, the IceTag data presented showed trends consistent with the both the calculated GPS speeds and the visual observation data. The lack of data due to malfunction, however, inhibited the depth of analysis able to be performed in this study. Accelerometer-based data in research are useful for research, identifying various behaviors such as grazing, resting, walking, and ruminating in livestock, and offer the ability to continuously monitor behavior remotely [5,42,43]. When used in terms of generating a ‘step count’ or level of acceleration, it is clear that though not as granular, GPS speeds can also produce a similar level of detail when accelerometers are absent. As represented in Figure 5, sheep in the IP spent more time walking across the day, in contrast to those in the NP, which had more distinct movement speeds in the morning and the afternoon across all trials; this was also reflected in their residency locations (Figure 7). The higher greenness of feed on offer in the NP consistently across the trials, compared to the IP (Figure 3), likely suggests that sheep in this paddock were spending less time grazing. Evidently, in T1 and T2 alone, sheep in the NP had higher counts of lying down than those in the IP (Figure 6c,d). Additionally, the decrease in the level of grazing activity for T3 in the IP (Figure 4c and Figure 5c) but increase in total pasture use (Figure 7c) could be due to having an increase in pasture quality in contrast to the prior trial/season, which inevitably led to a decrease in walking events to find quality pasture. This may be reflected with the known adaptivity animals have, where the level of foraging behavior changes depending on the available quantity and nutritive value of vegetation present [44].

### 4.2. Random Forest Models

Data from random forest models produced suggested that NDVI varied in its importance and significance for determining where sheep spent time in the paddock, across different seasons. Across the different trials, pasture quality, as identified through NDVI, was more variable for the IP than NP (Figure 3). Sheep in the IP evidently may be driven more by quality changes in the paddock than the sheep in the NP. Sheep are known to feed more selectively than other livestock and often do so to compensate for lower rumen residence time [14,45]. Whilst it is not possible to know exactly what was being consumed, pasture in the IP had clover species present (i.e., *Trifolum subterraneum*), which may have heavily influenced the selectivity by sheep when in the IP, due to the inherent preference known by livestock when grazing on clover species, for their ease to graze and high levels of sugars [44,46,47,48,49]. In contrast, the lack of such valued pasture species in the NP and lower levels of grazing activity across most trials for NP may further reiterate the narrative of sheep in IP spending more time looking for better-quality feed than what was often available (Figure 3, Figure 4, Figure 5 and Figure 7). The activity level differences between the paddocks may have attributed to the changes in liveweight seen, for example, wethers in the NP lost weight across the duration of the study. In addition to this, weight loss along with lower grazing activity could have also correlated with seasonal forage variation. Biomass production through pasture surveys was not explicitly measured in this study; through NDVI, it was evident that the production of ‘green’ forage was high in T1, decreased during the winter (T2) and increased again during T3 during the spring period (Figure 3a–c). Though these aforementioned green forage trends should correlate with weight changes [50], this was not the case for the study. As sheep in the IP were actively grazing on more nutritionally viable pasture, the lessened availability of such pasture in the NP impacted diet preferences of the grazing sheep, resulting in the consumption of less desirable pasture [14]. Evidently, sheep liveweight here was greater on the IP over the NP, consistent with previous research [51,52].

As previously discussed, EA and NO were ranked in the top three variables regardless of the treatment or trial period. This effect is likely due to the temporal correlation between GPS points [53]. Similarly, distance is a variable which can also be impacted by spatial autocorrelation between GPS points [53]. It has been stated in prior literature, however, that the position of water locations in the paddock can have an impact on the utilization, grazing, and overall distribution of livestock [3,12,19]. With the results evident, it is likely that NW was only significant and important in this study due to the paddock shape and the placement of the troughs over inherent spatial autocorrelation (Figure 2 and Figure 7). The literature has also suggested that in a larger pasture, livestock may head to water less often, spending more time away from it [54], which was a trend evident for both paddocks during all trials, as wethers spent most time at least 550 m away from the trough at the other end of the paddock (Figure A1 and Figure A2). A change was evident in T4, when residency was also high when wethers were within 50 m of water troughs, a likely event due to the warmer weather experienced in T4. The assumption here is that an increase in temperature led to an increase in water intake by sheep [55]. Warmer or cooler temperatures, high winds, and rainfall inherently also influenced location-based grazing for animals in this study. Primarily occurring during T2, weather variables R, WC, and T were ranked significantly higher for both paddocks than the rest of the trials. It is presumed that this is a large driving influence of grouping and residency changes in this trial in particular, as poor weather (high wind chill and rainfall) has been reported to impact heat loss from animals, influencing their location and utilization of shelters such as trees, shrubs, or structures within a landscape [56]. This was reflected in the trends evident in the residency by wethers within the paddock as during T2, wethers were often located near tree points which may have provided a form of shelter in both paddocks (Figure 7b). Additionally, any differences between LRI and the three weather variables during AH and GH could likely be attributed to the times which were included and excluded. For example, there is a lowered effect of weather variables on AH due to the larger range of hours included in contrast to the GH models, which saw higher influences of weather variables on LRI due to the smaller range of hours included (Figure A1 and Figure A2).

Landscape variables such as AS and EL in these models reflected that wethers here often preferred to spend more time in areas more northeast in the IP, whilst in the NP, animals spent the most time in the northern end (regardless of eastings), commonly in the areas where EL was the highest (Figure 7). This influence of topography has been reported in prior research, identifying the need of animals to be in areas of good visibility [57,58,59] and in areas where sun was located. It is presumed here that AS may have been further influenced by T during the trials. Furthermore, the differences in the presence of trees and shading between the two paddocks were substantial and may therefore have a large influence on grazing-based choices (Figure 7). As aforementioned, the NP had large amounts of tree cover in the northern end of the paddock, whilst the IP had a minimal line in the western border of the paddock (Figure 2). The presence of trees is known to influence pasture production on both native and introduced pastures [52,60,61], as well as the behavior of livestock [61]. Greater yields are evident when trees are not present; however, trees provide higher-quality pasture due to the access to nutrients trees inherently bring from soil nutrient dynamics [62,63]. Additionally, dry matter digestibility was found to be higher in pasture under tree cover [61,62]. Wethers in the IP were likely grazing closer to the areas where the trees were during all trials, due to more vegetation (as identified in higher NDVI values) present in the lower-lying areas, away from their camping location and closer to the tree line (Figure A1 and Figure 3). In contrast to this, wethers in the NP were found spending time away from tree lines (Figure A2) during T1 to T3. The comparison between grazing preferences in the two paddocks, aside from obvious tree cover differences, could have been attributed to the pasture itself. Despite nutrient concentration being higher in pastures under tree canopy, shade influences the quality and level of biomass more significantly in native pastures over those sown with introduced species [52,60]. Lastly, NF models and PDPs produced suggested that residency was greatest when closest to fence lines; however, it was not significant or important at all, regardless of the trial or the treatment. This effect is likely due to the camping locations predominantly being at the highest point of each paddock being directly next to a fence line (Figure 7).

### 4.3. Implications of Research and Future Directions

This research has highlighted that there are different drivers of grazing for sheep, depending on the pasture type (native vs. introduced) of the paddock, mainly the importance of NDVI for sheep in the IP over NP. As seen in the trends presented here, wethers on NP were often not walking as far as wethers in IP, which were presumably spending time searching for higher-quality pasture. Despite there being greater NDVI values in the NP than in the IP, the presence of valuable pasture species (i.e., clover) in areas of the IP as aforementioned is of note here. As indicated in the literature, the incorporation of more nutritional feed resources, such as improved pasture areas, is beneficial for covering nutritional requirements and maintaining sustainable production systems [44,64,65,66,67]. As native vegetation often has lower nutritive value and given the inherent grazing patterns observed in wethers in the IP spending more time grazing to find preferred plant species, it is suggested that graziers provide areas of these improved species amongst native pastures to meet the requirements of animals and provide herbage with higher digestibility and quality to assist in the maintenance of liveweights across seasons [52,67,68].

Recent research has often not addressed potential issues of spatial autocorrelation that can be inherent in GPS data, such as the work of Homburger et al. (2014), Gou et al. (2019), Brennan et al. (2020), and Pearson et al. (2021) [69,70,71,72]. In this study, we included EA and NO within the models as an attempt to determine the significance and importance of ‘space’ against data in a given point in time. While random forest models that explicitly consider the spatial autocorrelation of data are available (e.g., spatialRF), due to the size of the distance matrices that are produced for each pair of points, computing power can be a major limitation. In this study, as the size of the datasets used was quite large (~605,000 data points in total for AH, ~150,000 data points in total for GH), we were unable to implement spatial RFMs due to a lack of computing power. It is therefore suggested that future studies that focus on modelling GPS data of similar size to this utilize spatial RFM on high-performance computers to be able to sufficiently address this concern. Furthermore, it is suggested that additional research be undertaken to directly compare the various movement-based data with each other, to determine if modelling GPS behavior data is sufficient. This step is critical, as this type of research has the ability to allow for less labor-intensive monitoring of animals during tracking.

## 5. Conclusions

With the data presented here, it is evident that there are different drivers of grazing behaviors on pastures of native and improved species. As discussed previously, NDVI is often a significant variable for both paddocks; however, it can be suggested that the quality of pasture was more important for wethers in paddocks of improved species than it was for wethers on native species. The results discussed in this study highlight the ability to use predictive models such as RFMs to provide valuable insights on behavior-based modelling of GPS data. With data such as these, there is greater ability to further enhance current knowledge on the impacts of multiple variables on grazing preferences to assist in developing management strategies to improve the sustainable grazing of sheep within Australia.

## Figures and Tables

**Figure 1 animals-13-01500-f001:**
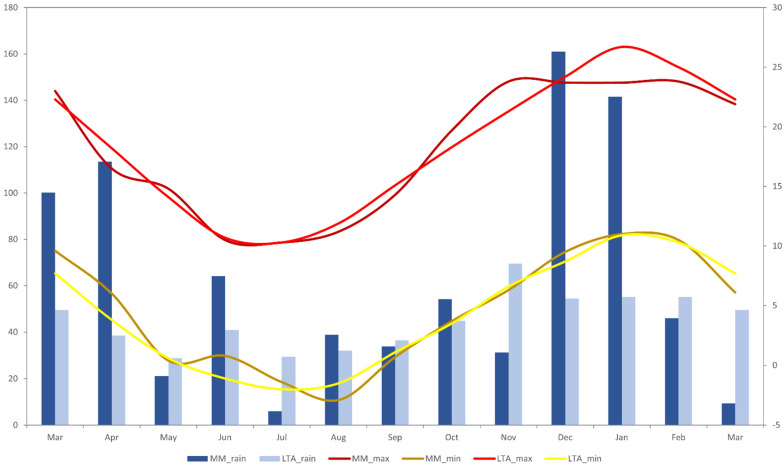
Comparison of long-term average temperature and rainfall data (1976–2023) and monthly average temperature and rainfall data (2014–2015) for the Coolringdon site over the period of the study [23]. MM = mean monthly; LTA = long-term average.

**Figure 2 animals-13-01500-f002:**
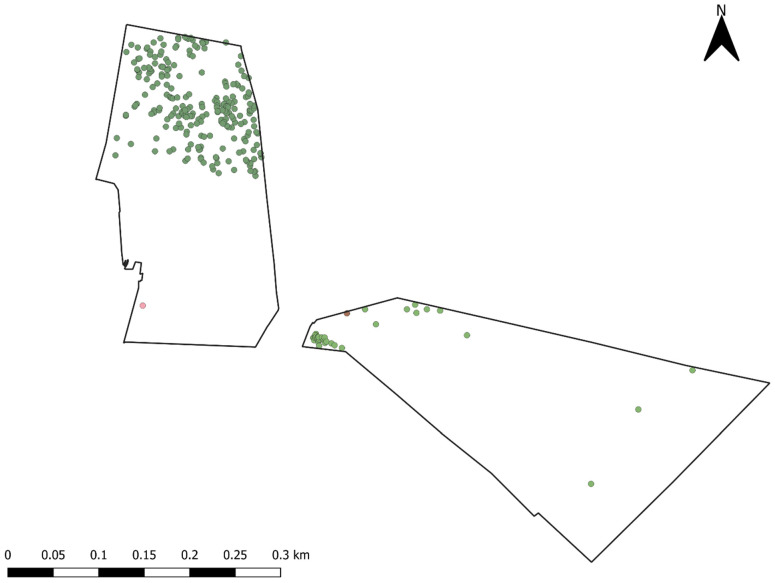
Map of native and improved paddocks used for the duration of the study and selected features. Red circles = water troughs; Green circles = GPS-logged tree locations.

**Figure 3 animals-13-01500-f003:**
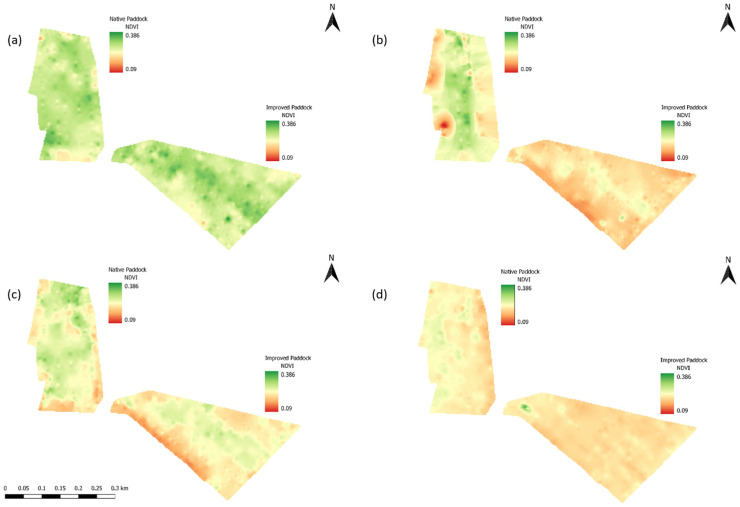
Normalized Difference Vegetation Index (NDVI) maps of native and improved paddocks during (**a**) Trial 1 (23–28 April 2014), (**b**) Trial 2 (9–14 July 2014), (**c**) Trial 3 (11–16 November 2014), and (**d**) Trial 4 (27 March–1 April 2015).

**Figure 4 animals-13-01500-f004:**
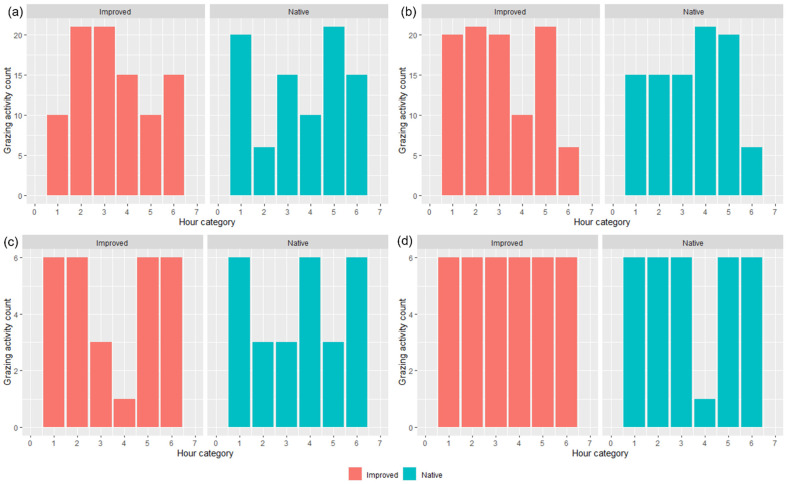
Average (*n* = 15 bibbed sheep) grazing activity counts for both native and improved paddocks during (**a**) Trial 1, (**b**) Trial 2, (**c**) Trial 3, and (**d**) Trial 4, per hour. NB: due to the slightly different start times across the trials (i.e., 7:30 a.m. vs. 6:00 a.m. start; effects of daylight savings), hour is presented in categorical notion whereby ‘1′ was the first hour of recording, ‘2′ was the second, and so on.

**Figure 5 animals-13-01500-f005:**
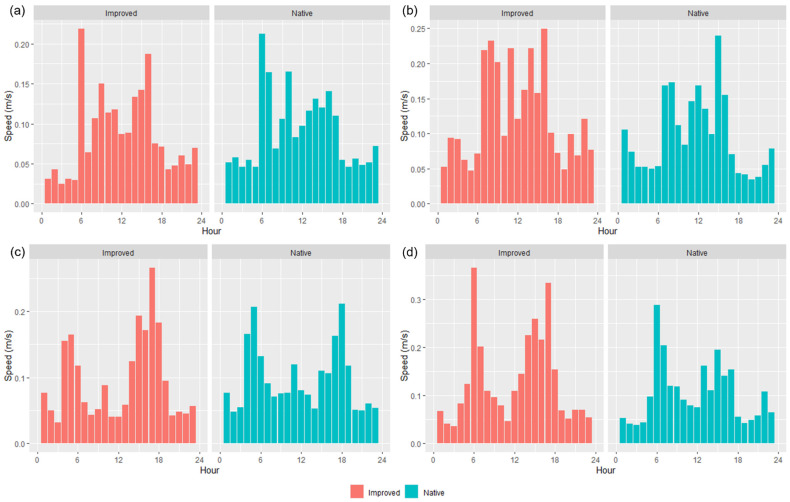
Calculated total average (*n* = 15 collared sheep) speed (m/s) of sheep using GPS collars, for both native and improved paddocks during (**a**) Trial 1, (**b**) Trial 2, (**c**) Trial 3, and (**d**) Trial 4.

**Figure 6 animals-13-01500-f006:**
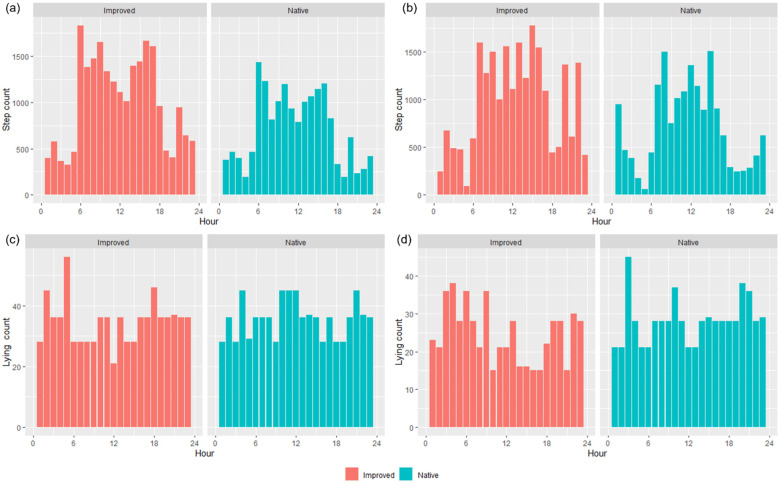
IceTag data for the average step count (**a**,**b**) and average lying count (**c**,**d**) per hour for Trial 1 (**a**,**c**) and Trial 2 (**b**,**d**) in both the improved and native paddocks.

**Figure 7 animals-13-01500-f007:**
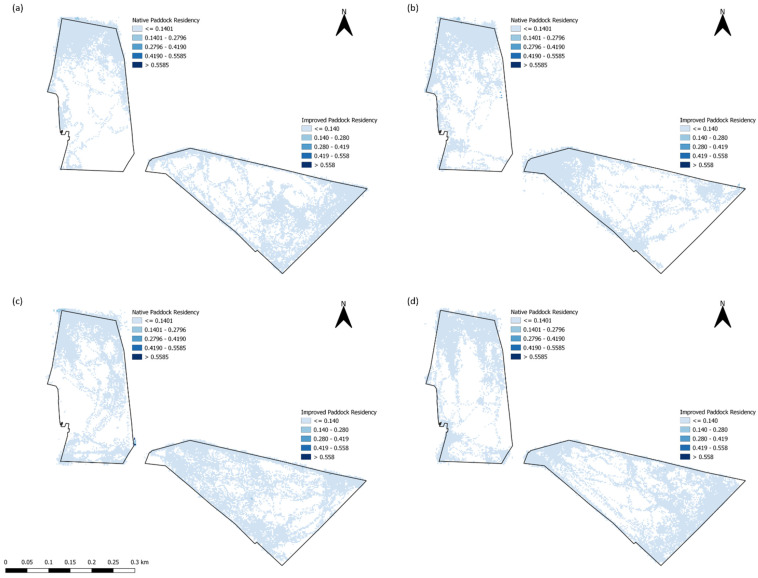
Livestock residency index (LRI) values in the improved and native paddocks for (**a**) Trial 1, (**b**) Trial 2, (**c**) Trial 3, and (**d**) Trial 4.

**Table 1 animals-13-01500-t001:** Model performance of OOB R^2^ and MSE values for the models produced for each trial, treatment, and data type—all hours (AH) and grazing hours (GH).

	**Trial 1**				**Trial 2**			
	Improved		Native		Improved		Native	
	AH	GH	AH	GH	AH	GH	SH	GH
OOB R^2^	0.978	0.982	0.949	0.957	0.958	0.952	0.944	0.956
MSE	2.0 × 10^−6^	2.0 × 10^−6^	3.0 × 10^−6^	3.0 × 10^−6^	3.0 × 10^−6^	3.0 × 10^−6^	3.0 × 10^−6^	4.0 × 10^−6^
	**Trial 3**				**Trial 4**			
	Improved		Native		Improved		Native	
	AH	GH	AH	GH	AH	GH	Ah	GH
OOB R^2^	0.981	0.977	0.941	0.943	0.976	0.975	0.949	0.950
MSE	2.0 × 10^−6^	2.0 × 10^−6^	3.0 × 10^−6^	3.0 × 10^−6^	2.0 × 10^−6^	2.0 × 10^−6^	3.0 × 10^−6^	3.0 × 10^−6^

**Table 2 animals-13-01500-t002:** Rankings for variables in models produced based on importance value for each trial, treatment, and data type—all hours (AH), and grazing hours (GH).

	Trial 1				Trial 2				Trial 3				Trial 4			
	Improved		Native		Improved		Native		Improved		Native		Improved		Native	
Variable	AH	GH	AH	GH	AH	GH	AH	GH	AH	GH	AH	GH	AH	GH	AH	GH
Animal (A)	11th	10th	12th	11th	10th *	10th *	12th	12th	11th	12th	12th *	12th	10th *	10th *	11th	12th
Eastings (EA)	4th	4th	1st *	1st *	3rd	1st	1st	2nd	1st *	1st *	2nd *	2nd *	2nd *	2nd *	2nd *	1st *
Northings (NO)	1st	1st *	3rd	2nd *	1st	4th	3rd	5th	2nd *	4th	3rd	3rd *	1st *	1st *	1st *	3rd *
Near distance to trees (NT)	8th	6th *	7th	6th *	7th	7th	7th	7th	5th *	8th *	6th *	5th *	3rd *	8th	7th	5th *
Near distance to water (NW)	6th	7th	2nd	3rd *	4th	3rd	2nd	4th	3rd	2nd *	1st *	1st *	4th	3rd	3rd	2nd *
Near distance to fences (NF)	12th	11th	10th	10th	12th	12th	11th	11th	12th	11th	11th	11th	12th	12th	10th	11th
NDVI	2nd *	2nd *	8th *	7th *	8th	8th	8th	8th	4th *	3rd *	8th	8th *	6th *	4th *	8th	8th *
Aspect (AS)	9th	9th	9th	8th *	11th	11th	9th	9th	9th	9th	9th	9th *	9th	9th	9th *	9th *
Elevation (EL)	5th	5th	5th	4th *	6th	6th	5th	3rd	6th	7th	4th	4th *	7th	5th	5th	4th *
Temperature (T)	3rd *	3rd *	4th *	5th *	2nd *	2nd *	6th *	6th *	7th *	6th *	5th *	6th *	5th *	6th *	4th *	6th *
Rainfall (R)	10th *	12th	11th *	12th *	9th *	9th *	10th *	10th *	10th *	10th *	10th *	10th *	11th	11th *	12th	10th *
Wind Chill (WC)	7th *	8th	6th *	9th *	5th *	5th *	4th *	1st *	8th *	5th *	7th *	7th	8th *	7th *	6th *	7th

* Indicates significance (*p* < 0.01).

## Data Availability

The data which support the findings in this study are available from the corresponding author upon reasonable request.

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
