# Peer review of "Exploring ‘Wether’ Grazing Patterns Differed in Native or Introduced Pastures in the Monaro Region of Australia"

_animals, 2023, doi:10.3390/ani13091500_

Round 1
Reviewer 1 Report
Dear authors. It was a complex and well-conducted work
Congratulations
The specific number of animals used for GPS and other tracking devices needs to be clearly defined in sections such as abstract and methodologies. Also, (i) number of observed animals, (ii) the methodology to select those animals.
Authors clearly stated that the objective was to determine variables influencing sheep location. But for some readers the differentiation between nature and improve pastures could be seen as a comparison. Thus, my recommendation is to clarify within the discussion that this work was not a comparison between two pasture-types.
The LRI methodology was important in the present work. Nonetheless, there are little references to pioneer works. Authors are invited to include more information in this topic
Although original and clear as a “word game”, the title could be modified. It could be somewhat difficult to interpret in the original way proposed by the authors
The paragraph between L82-92 is more related with the material and methods section
Some ideas within the discussion assumed the consumption of specific resources (e.g. Trifolium). However, the nature of GPS and tracking devices do not allow to effectively determine the specific plant resources consumed by wethers. A consideration of this topic might reinforce the discussion
A more detailed discussion on the NDVI and its importance in livestock (sheep for the sake of the present work) production is suggested.
____
Abstract: The acronyms for GPS and NDVI were not defined yet
L46-48: Please add a reference supporting the well-known fact of high input labour in observational methods
L84: There is a space between “often” and “an”
L98: NDVI was not defined yet
L111: Pease define the seasons
Legends in figure 1 are not self-sufficient. Please consider modify
L132: Is this presentation order of plant resources related with availability? If not, please arrange them alphabetically
L135: Scientific name of pine trees please
L152: Acronyms for VESPER were not defined yet
L161: Pease write completely “approximate”
Author Response
Thank you for taking the time to review the manuscript. We have reviewed your line by line comments and made amendments per your suggestions. For each of your points you have presented with detail, we have provided information on changes made and a few comments in reply.
Point 1: The specific number of animals used for GPS and other tracking devices needs to be clearly defined in sections such as abstract and methodologies. Also, (i) number of observed animals, (ii) the methodology to select those animals.
Additional sentences and a re-edited paragraph have been provided in the methodology section highlighting the number of collars, bibs, and accelerometers that were fitted on the animals, for clarity. Minor edit made in the abstract for flow. No other additions were added in the abstract due to word count limitations. The animals observed were randomly selected as well during the monitoring process.
Point 2: Authors clearly stated that the objective was to determine variables influencing sheep location. But for some readers the differentiation between nature and improve pastures could be seen as a comparison. Thus, my recommendation is to clarify within the discussion that this work was not a comparison between two pasture-types.
Objectives were to determine the variables on both native and improved pastures, as they were inherently different from each other. Some sentences have been added in the introduction and discussion to address this comment. It has been highlighted throughout the text that there are differences (thus, implying a comparison between the two).
Point 3: The LRI methodology was important in the present work. Nonetheless, there are little references to pioneer works. Authors are invited to include more information in this topic
Added references from the work of Trotter et al., 2010, 2008, to encapsulate this. Please note, this work predominantly focused on the value obtained (i.e., livestock residency index value) whereas work by others which have generated maps based on LRI.
Point 4: Although original and clear as a “word game”, the title could be modified. It could be somewhat difficult to interpret in the original way proposed by the authors
The authors liked the idea of using word play in the title as a fun approach towards science. We have included the description of a wether in the abstract (closer to where the title is) which may have reduced any concerns with clarity if this is the key issue.
Point 5: The paragraph between L82-92 is more related with the material and methods section
Amendments have been done in text in the introduction and the methods to address this comment.
Point 6: Some ideas within the discussion assumed the consumption of specific resources (e.g. Trifolium). However, the nature of GPS and tracking devices do not allow to effectively determine the specific plant resources consumed by wethers. A consideration of this topic might reinforce the discussion
Discussion has been edited to include this and provide more clarity.
Point 7: A more detailed discussion on the NDVI and its importance in livestock (sheep for the sake of the present work) production is suggested
Additional line included in the methods to discuss its importance.
Thank you once again for providing clear constructive feedback on this manuscript. We hope that any comments raised have been adequately addressed by us here.
Reviewer 2 Report
Thank you for letting me review this interesting manuscript. I recommend its publication after minor revision. Please, pay attention that results can be difficult to follow because of the large use of abbreviations.
The main question addressed by the research is to understand what potential variables are significant for predicting where sheep spent the most time in native (NP) and improved (IP) paddocks. The topic is relevant in the field and addresses a specific gap in the knowledge of livestock management. The study focuses on using GPS tracking collars to monitor the behaviour of sheep in different types of paddocks and identify the variables that are most significant in predicting where the sheep spend their time. This information can help improve the management of livestock for optimal production. The study also highlights the potential of predictive models to provide insights into animal behaviour based on GPS data, which is a relatively new and rapidly developing area in the field of livestock management. Therefore, the research is both original and relevant in the field. This research adds to the subject area of livestock management by using GPS tracking collars to monitor the behaviour of sheep in different types of paddocks and identifying the variables that are most significant in predicting where the sheep spend their time. While similar studies have been conducted in the past, this research is unique in its focus on the specific factors that influence the behaviour of sheep in the different types of paddocks. Additionally, the study highlights the potential of predictive models to provide insights into animal behaviour based on GPS data, which is a relatively new and rapidly developing area in the field of livestock management. Therefore, this research adds new insights and knowledge to the subject area compared with other published material.
The study has a strong methodology and provides valuable insights into the behaviour of sheep in different types of paddocks. However, there are some improvements that the authors could consider to strengthen the methodology:
Sample Size: The study used a relatively small sample size of 15 sheep in the IP and 27 sheep in the NP. A larger sample size would provide more robust results and increase the generalizability of the findings.
Duration of Trials: The study only conducted trials over four six-day periods in different seasons. It would be beneficial to extend the duration of the trials to include a longer time period and more seasons to account for potential seasonal variations in the behaviour of sheep.
Replication: The study was conducted on a single property in Southern New South Wales, Australia. Replication of the study in different locations would provide further validation of the findings and ensure that they are not specific to this particular property. Further controls that the authors could consider include: Accounting for individual differences: The study did not account for individual differences in sheep behaviour. Future studies could consider tracking individual sheep to account for any differences in behaviour between individuals.
Controlling for environmental factors: The study identified several environmental factors that were significant in predicting sheep behaviour, but it did not control for these factors. Future studies could consider controlling for environmental factors such as weather conditions and vegetation cover to isolate the effects of other variables on sheep behaviour.
Overall, the authors have conducted a well-designed study, but these improvements and controls could help strengthen the methodology and provide more robust results.
The conclusions of the study are consistent with the evidence and arguments presented, and they address the main question posed by the research. The study aimed to understand what potential variables are significant for predicting where sheep spent the most time in native (NP) and improved (IP) paddocks and the results showed that NDVI was significant for wethers in the IP, but not the NP, suggesting that quality of pasture was key for wethers in the IP. Elevation, temperature, and near distance to trees were important and significant for predicting residency of wethers in both the IP and NP. The conclusions of the study are based on the analysis of data collected from GPS tracking collars on sheep in different types of paddocks, and the authors provide a clear and logical argument for how the results support their conclusions. Therefore, the conclusions are consistent with the evidence and arguments presented, and they effectively address the main question posed by the research.
My additional comments are in the attached file.

Author Response
Thank you for taking the time to review the manuscript! We greatly appreciate the amount of detail you have provided in your feedback, and the level of granularity you have provided in your attached file. I have addressed all comments and made amendments as suggested per your attached file. We would like to acknowledge your concerns about the heavy use of abbreviations, however we felt that it would end up sounding quite repetitive and be a longer text to read without the abbreviations.
Point 1 and 3: Sample Size: The study used a relatively small sample size of 15 sheep in the IP and 27 sheep in the NP. A larger sample size would provide more robust results and increase the generalizability of the findings. Replication: The study was conducted on a single property in Southern New South Wales, Australia. Replication of the study in different locations would provide further validation of the findings and ensure that they are not specific to this particular property.
The sample size of the study utilised flocks of 25 sheep per paddock (25 in the IP, 25 in the NP, 50 total), and had 15 sheep per paddock with GPS collars and bibs, whilst the remaining 10 were buffer animals. I have amended the section in which this is discussed in case there were any misunderstandings due to the lack of clarity there. Any larger number of sheep in the mob may not have necessarily provided any differences in the general behaviour as by this size, they are likely to be enacting as a mob as well.
On a relevant note, I have previously published a study (Parnell et al., 2022) using similar methods to understand grazing patterns exhibited by wethers in Orange, New South Wales, whereby the study utilised five sheep (as a ‘high stocking rate’) in a set of three of replicate paddocks, and three sheep (as a ‘low stocking rate’) in the other set of replicate paddocks that were ~0.5 ha. The results between the two studies are immensely different based on the model performance alone. No comparisons are done between the two as the setup of each of the studies did vary, but the authors agree that replication at another location may be beneficial to further validate the findings.
Point 2: Duration of Trials: The study only conducted trials over four six-day periods in different seasons. It would be beneficial to extend the duration of the trials to include a longer time period and more seasons to account for potential seasonal variations in the behaviour of sheep.
Though we agree that the study would have greatly benefitted from observations and tracking occurring for greater periods of time to obtain further granularity on seasonal effects, this is no longer plausible due to when the data collection was performed (2014). In addition, during this time, there were constraints with the technology used (IceTag accelerometers) and their battery life which hindered the possible use for prolonged periods (and explains the lack of data for trials 3 and 4), as well as the inability to have an observer physically out watching the sheep for the duration of the year. However, we do have data for six days during different seasonal periods as Trial 1 occurred during Australia’s autumn (April 2014), trial 2 for winter (July 2014), trial 3 for spring (November 2014) and trial 4 summer/autumn (March 2015), which we believe may at least encapsulate a small portion of the variation.
Point 4: Further controls that the authors could consider include: Accounting for individual differences: The study did not account for individual differences in sheep behaviour. Future studies could consider tracking individual sheep to account for any differences in behaviour between individuals.
Regarding your comment on individual sheep behaviour, I have clarified the inclusion of animal (A; as highlighted in your attached file) in the model/paper since it may have been removed as an oversight in initial edits. We included A in the model as a metric to identify if there were any significant influences by individual animal behaviour. With the partial dependence plots (PDP) produced in the appendix figures, you can identify which animals often had higher levels of residency than others (suggesting they stayed in similar places). However, as highlighted in the results, A was not significant, and was often always ranked last, so the discussion of this was removed (mainly to save on space). Furthermore, from the observational data, each of the animals that were being tracked had identifying bibs, and often, these animals had similar behaviours/patterns. There were no clear differences in individual behaviour in the animals.
Point 5: Controlling for environmental factors: The study identified several environmental factors that were significant in predicting sheep behaviour, but it did not control for these factors. Future studies could consider controlling for environmental factors such as weather conditions and vegetation cover to isolate the effects of other variables on sheep behaviour.
We included weather variables such as rainfall, temperature, and calculated sheep chill index (discussed in text) to encapsulate the impact these variables may have in location based grazing choices. Interestingly (and as discussed) there was a long summer drought prior to the start of the study and an unusual pattern of rainfall during the study, which may have influenced pasture production positively for when this study was undertaken.
A different site entirely would have needed to have been picked to be able to control vegetation cover/offer consistency between the two paddocks, however these two paddocks were chosen for their closeness and similarities in size, and the two distinct pasture types on offer which were the main priorities when sample design occurred.
Thank you once again for reading this manuscript and taking the time to provide valuable, constructive feedback. I hope that our reply to your comments make sense and provide additional clarity.